# Channel Characterization and Modeling for 6G UAV-Assisted Emergency Communications in Complicated Mountainous Scenarios

**DOI:** 10.3390/s23114998

**Published:** 2023-05-23

**Authors:** Zhaolei Zhang, Yu Liu, Jie Huang, Jingfan Zhang, Jingquan Li, Ruisi He

**Affiliations:** 1School of Microelectronics, Shandong University, Jinan 250101, China; zhaoleizhang@mail.sdu.edu.cn (Z.Z.); 202132424@mail.sdu.edu.cn (J.Z.); 202232357@mail.sdu.edu.cn (J.L.); 2The State Key Laboratory of Rail Traffic Control and Safety, Beijing Jiaotong University, Beijing 100044, China; 3The National Mobile Communications Research Laboratory, School of Information Science and Engineering, Southeast University, Nanjing 210096, China; j_huang@seu.edu.cn; 4The Purple Mountain Laboratories, Nanjing 211111, China

**Keywords:** channel characteristics, millimeter wave, 6G, UAV-assisted emergency communications, mountainous scenarios

## Abstract

Regarding the new demands and challenges of sixth-generation (6G) mobile communications, wireless networks are undergoing a significant shift from traditional terrestrial networks to space-air-ground-sea-integrated networks. Unmanned aerial vehicle (UAV) communications in complicated mountainous scenarios are typical applications and have practical implications, especially in emergency communications. In this paper, the ray-tracing (RT) method was applied to reconstruct the propagation scenario and then acquire the wireless channel data. Channel measurements are also conducted in real mountainous scenarios for verification. By setting different flight positions, trajectories, and altitudes, channel data in the millimeter wave (mmWave) band was obtained. Important statistical properties, such as the power delay profile (PDP), Rician K-factor, path loss (PL), root mean square (RMS) delay spread (DS), RMS angular spreads (ASs), and channel capacity were compared and analyzed. The effects of different frequency bands on channel characteristics at 3.5 GHz, 4.9 GHz, 28 GHz, and 38 GHz bands in mountainous scenarios were considered. Furthermore, the effects of extreme weather, especially different precipitation, on the channel characteristics were analyzed. The related results can provide fundamental support for the design and performance evaluation of future 6G UAV-assisted sensor networks in complicated mountainous scenarios.

## 1. Introduction

In the vision of future sixth-generation (6G) space-air-ground-sea-integrated networks, unmanned aerial vehicle (UAV) communications can greatly extend the technical boundaries of mobile wireless communications [1,2,3]. Meanwhile, emergency communications in complicated mountainous scenarios have drawn wide concern and become a hot research topic. Due to the high mobility of the UAV, it can be applied to establish emergency communication services that guarantee ground sensor nodes and users in the first instance [4]. Most existing research on UAV communications focuses on campus scenes, urban scenes, etc., and few focus on mountainous scenarios with potential practical significance. In complicated mountainous scenarios during emergency communications, UAV-assisted sensor networks can achieve fire warnings, rescue missions, reconnaissance, detection, etc., and serve as communication relays [5,6,7,8]. It is well known that the development, verification, and evaluation of wireless sensor communication networks are all dependent on accurate channel models [9]. Compared to traditional terrestrial communications, UAV communications in complicated mountainous scenarios have unique channel characteristics, such as blocking of the mountain, dense scatterer environments, three-dimensional (3D) arbitrary trajectories, and fast-changing UAV nodes [9,10]. Therefore, it is crucial to have an in-depth understanding of channel characteristics for UAV-assisted emergency communication systems in complicated mountainous scenarios.

Research on UAV channel modeling can be divided into two main approaches, i.e., measurement-based empirical channel models and simulation-based deterministic or stochastic channel models. Channel measurements generally adopt either frequency domain channel measurement or time domain channel measurement [11,12,13,14]. In [15], various flight trajectories in hilly scenarios were measured based on the time domain channel measurement system, and some key channel characteristics of the UAV air-to-ground (AG) channel were analyzed and compared. However, it focused on the sub-6 GHz frequency band and only considered hilly rather than mountainous scenarios. In [16], AG channel measurements were performed in a hilly environment with 500 MHz bandwidth at 6.5 GHz. In [17], UAV AG channels were measured at 968 MHz and 5 GHz in the hilly areas near Latrobe, PA, USA, and Palmdale, CA, USA, as well as in the mountainous environment near Telluride, Colorado. Besides, channel measurements were extended to suburban and near-urban environments in [18]. Moreover, a large inflexible aircraft was required to carry equipment during channel measurements [17,18]. It was not suitable for low altitude UAV communications and did not involve millimeter wave (mmWave) technology. In [19], channel measurements at the 28 GHz frequency band were carried out in residential and mountainous desert terrains, but the propagation distance was limited. Due to the complex environment, difficult experimental conditions, and expensive experimental equipment, there are fewer UAV mmWave channel measurements, especially in complicated mountainous scenarios.

In the deterministic channel modeling for UAV communications, the ray-tracing (RT) method demonstrates excellent performance and can take full account of environmental conditions. In [20], the propagation characteristics at 28 GHz in dense urban, suburban, rural, and maritime scenarios were analyzed using the RT method. It was found that the multipath components (MPCs) arriving at the receiver (Rx) presented birth and death phenomena along the UAV flight trajectory. In [21], the channel characteristics of a suburban open environment were analyzed using the RT method. In [22], an RT-based Monte-Carlo method was conducted to analyze and evaluate UAV AG channel model performance. In stochastic channel modeling, geometry-based stochastic models (GBSMs) have been widely used, and the corresponding channel characteristics have been analyzed [23,24,25,26,27,28,29,30,31,32]. In [25], a novel non-stationary maritime UAV-to-ship GBSM was proposed. Massive multiple-input multiple-output (MIMO) channel models based on regular and irregular shaped GBSMs were introduced in [26,27]. The motion characteristics of arbitrary 3D trajectories of UAVs were considered, and time-varying parameters were used to simulate the movement of the transmitter (Tx), Rx, and scatterers. In [28], a 3D non-stationary GBSM for a UAV-to-vehicle mmWave beamforming channel was proposed. Based on an angular estimation algorithm, a novel UAV channel model for UAV AG communications was proposed in [29]. In [30], a hybrid hopping-based channel model was proposed to improve the model accuracy. In addition, time–space consistency and the impacts of UAV posture variation and fuselage scattering were considered in [31,32]. However, both channel models were more suitable for urban scenarios and did not take into account the uniqueness of the mountain scene. In [33], the UAV AG channel model was calculated through the geometric relationship of the ramps, but only for a simple single mountain scenario. In addition, some standard UAV communication channel models have been proposed, such as the third generation partnership project (3GPP) standard in [34]. However, few models have taken complicated mountainous scenarios into account.

As mountainous scenarios are specific, the deterministic RT method is more relevant to real environments. The relevant channel characteristics are analyzed using the RT method. In addition, in the event of devastating disasters or emergencies, emergency communication networks face complex and variable terrain and climatic environments, which bring new challenges to UAV communications. Therefore, it is essential to investigate the key channel characteristics of UAV communications in mountainous scenarios, which can provide a reference for the design and evaluation of future UAV communications in complicated mountainous scenarios. The main contributions are listed as follows:The UAV-assisted emergency communications network architecture in the mountainous scenarios was built based on a realistic scenario using the RT method. Meanwhile, the channel measurement was carried out in mountainous scenarios. Using the measurement data, the accuracy and practicality of the RT method could be well verified.Based on the channel data using the RT method, the effects of a UAV navigating mountain positions on channel characteristics were derived and analyzed. Also, different influences of flight altitudes and common flight trajectories were compared.The channel propagation characteristics of different frequency bands (3.5 GHz, 4.9 GHz, 28 GHz, and 38 GHz) were analyzed and compared in complicated mountainous scenarios, e.g., Rician K-factor, path loss (PL), the angle domain characteristics, the delay domain characteristics, channel capacity, etc. Furthermore, the effect of severe weather such as rain on propagation was simulated and studied.

The rest of this paper is organized as follows. Section 2 proposes the UAV fundamental network framework in complicated mountainous scenarios and gives simulation procedures. The RT-based typical channel characteristics are presented in Section 3. Section 4 gives the relevant numerical analysis of statistical properties, which were verified by measured data. Finally, conclusions are drawn in Section 5.

## 2. UAV Communications Network Architecture and Simulation Setup

### 2.1. Descriptions of Network Architecture

A brief description of the UAV-assisted emergency communications network architecture in complicated mountainous scenarios is shown in Figure 1. The UAV side is set as Tx, and the ground side is set to Rx. Due to the complex terrain, mountains of various morphologies can bring more reflections and cause multipath fading. The UAV AG channel modeling in complicated mountainous scenarios needs to consider all possible multipath types. Here, the entire communications link contains three components: The line-of-sight (LoS) component represents the UAV communications directly with the ground end, and the LoS component usually has the largest power. The single bounce (SB) component means the received signal is reflected once to reach the ground end. In mountainous scenarios, the SB generally divides into two categories—one is the scattering path due to ground reflections near the Rx, and the other is from the mountains. The multiple bounce (MB) component indicates that the received signal experiences the effects of multiple environmental characteristics during propagation, e.g., it reaches the Rx end after one reflection from a mountainous area and then another reflection from the ground. The MB power is often low and not easily detected by the Rx.

### 2.2. Mountainous Scenarios Reconstruction

The RT method is a deterministic channel modeling approach and has been used in numerous studies and analyses, especially when combined with electromagnetic wave propagation to calculate characteristics. In this paper, RT-based Wireless Insite (WI) software (version number: 3.3; creator: REMCOM; location: Nanjing, China) was used to simulate UAV AG communications in mountainous scenarios [35]. Reconstructions were conducted based on realistic scenarios located in the southern mountainous region of Jinan, Shandong Province, west of the Ji-Tai highway and north of the S317 road. The specific range is 117.0601° E to 117.1081° E and 36.4993° N to 36.5210° N. The terrain of the simulation environment is shown in Figure 2. The selected terrain material is wet earth. The characteristics of the terrain material parameters tend to be varied at different frequency bands, as shown in Table 1.

In order to simulate the mountainous scenarios more precisely, vegetation information was also considered. The vegetation data from the Global Land Cover Characteristics (GLCC) satellite database [36] was imported, and the vegetation was covered on the specific areas according to the information obtained from the satellite. The vegetation contains deciduous broadleaf woodland, cropland, grass, etc. The specific physical parameters of trees and vegetation are referred from the International Telecommunication Union (ITU) database and WI software document [35,37], e.g., the radius; thickness and density of the leaf, branch, and blade; conductivity; attenuation; etc. In addition, in the software, the vegetation categories were selected as BioPhysical. The diffuse reflections were also introduced to suit the actual propagation. The lambertian diffuse reflection model was chosen, and the scattering factors varied with frequency bands (scattering factors were 0.400, 0.400, 0.410, and 0.484 for 3.5 GHz, 4.9 GHz, 28 GHz, and 38 GHz, respectively). The atmospheric components were taken into account, such as temperature, atmospheric pressure, and humidity. The effects of extreme weather such as rain were also simulated. The atmospheric compositions were modified, and post-processing was applied to each ray, which can simulate the effect of rain on the channel. The formula details will be mentioned in the Section 4.4.

The type of UAV considered in this paper is a rotorcraft, which is small in size and has small wings, so the UAV-carried Tx was considered as an emission source point in order to be effective and not lose generality. In addition, the electromagnetic rays emitted by the UAV radiate in all directions, and the rays hit obstacles, i.e., mountains, ground, trees, etc. Reflection and bypassing phenomena often occur, and generally no refraction or projection occurs. The simulation parameters of the UAV have an important influence on the signal propagation. From the issues often considered in UAV-assisted emergency communications in real mountainous scenarios, this paper explored the impacts of the following problems on the UAV channel characteristics: (**I**) UAV flight position; (**II**) UAV flight trajectory and altitude; and (**III**) UAV carrier frequency and environmental factors. All the above mentioned issues can provide a reference for the design and optimization of future UAV communication systems. Figure 3 shows the satellite view. The detailed flight and simulation parameters are given in Table 2.

## 3. Ray-Tracing Based Typical Channel Characterization

### 3.1. Channel Data Acquisition

By using the RT method, the amplitude, phase, and delay data of each MPC can be obtained. In the *i*-th MPC, it can be uniformly expressed as $\alpha_{i}$, $\phi_{i}$, and $\tau_{i}$, respectively. The channel impulse response (CIR) is acquired to mimic the UAV AG channel and is calculated using the inverse Fourier transform of channel transfer function (CTF). The CTF is expressed as [38]
(1)$$H\left(f_{k},t\right)=\sum_{i=1}^{I(t)}\alpha_{i}\left(t\right)e^{j\phi_{i}\left(t\right)}e^{-j2\pi f_{k}\tau_{i}\left(t\right)},$$
where the MPC number is denoted as *I*. All these parameters are time-variant. The *k*-th frequency sampling point is denoted $f_{k}$. These sampling points together form the $H\left(f,t\right)$ complex matrix. Using the CTF, the CIR is calculated as
(2)$$h\left(t,\tau \right)=\mathrm{IFFT}\left(H\left(f,t\right)\right),$$
where τ and *f* are Fourier transform pairs. The time-variant characteristics of CIR stem from the time evolution of the scattering clusters.

### 3.2. Typical Channel Characterization

#### 3.2.1. Power Delay Profile

The PDP reflects power distribution in the delay domain and consists of the delay τ and the power *P*. The PDP is calculated as
(3)$$\Upsilon \left(t,\tau \right)={\left|h\left(t,\tau \right)\right|}^{2}.$$

Different UAV flight speeds, trajectories, altitudes, and the locations of mountainous scenarios will lead to different trends in the PDP. If *t* is fixed, the only variable in Equation (Equation 3) is τ, which reflects the delay distribution at a particular location or moment.

#### 3.2.2. Path Loss and Shadow Fading

The PL is an important large scale parameter in channel characterization and is used to describe the propagation loss. It can be defined as
(4)$$\Gamma \left(\mathrm{dB}\right)=P_{r}-P_{t}-G_{t}-G_{r},$$
where Γ is PL, $P_{t}$ denotes transmitted power, and $P_{r}$ represents received power. The gains of transmit and receive antennas are $G_{r}$ and $G_{t}$, respectively. After the PL is known, the floating intercept (FI) model is commonly fitted to PL data [12]. It can be calculated as
(5)$$\Gamma \left(d\right)=10n\log_{10}\left(d\right)+\nu +X_{s},$$
where *n* is the PL exponent (PLE), ν denotes the intercept, and *d* is the 3D straight-line distance from the Tx to Rx in m unit. SF is caused by obstacles between the Tx and Rx and tends to obey a Gaussian distribution with a zero mean and standard deviation of σ. Therefore, the fit is generally carried out with three parameters: *n*, ν, and σ.

#### 3.2.3. Rician K-Factor

The Rician K-factor reflects the quality of the UAV communications and represents the strength of the LoS component during signal transmission [15]. It is defined as
(6)$$K\left(\mathrm{dB}\right)=\frac{P^{LoS}}{\sum_{i=1}^{I}P_{i}-P^{LoS}},$$
where $P_{i}$ denotes the *i*-th path power, and $P^{LoS}$ denotes the power value of the LoS ray.

#### 3.2.4. The Delay Spread and Angular Spread

The root mean square (RMS) delay spread (DS) characterizes the channel time dispersion and reflects the magnitude of the multipath effects. The RMS DS is an essential small scale parameter and is calculated as
(7)$$\sigma_{\tau }=\sqrt{\frac{\sum_{i=1}^{I}P\left(\tau_{i}\right)\times \tau_{i}^{2}}{\sum_{i=1}^{I}P(\tau_{i})}-{\left(\frac{\sum_{i=1}^{I}P\left(\tau_{i}\right)\times \tau_{i}}{\sum_{i=1}^{I}P(\tau_{i})}\right)}^{2}},$$
where $\tau_{i}$ and $P(\tau_{i})$ are the delay and power of the *i*-th MPC ray, respectively. The RMS angular spreads (ASs) also characterize the channel angular dispersion characterization, which can be denoted by
(8)$$\sigma_{\theta }=\sqrt{\frac{\sum_{i=1}^{I}P\left(\theta_{i}\right)\times {\theta_{i}}^{2}}{\sum_{i=1}^{I}P\left(\theta_{i}\right)}-{\left(\frac{\sum_{i=1}^{I}P(\theta_{i})\times \theta_{i}}{\sum_{i=1}^{I}P(\theta_{i})}\right)}^{2}}$$
(9)$$\sigma_{\varphi }=\sqrt{\frac{\sum_{i=1}^{I}P\left(\varphi_{i}\right)\times {\varphi_{i}}^{2}}{\sum_{i=1}^{I}P\left(\varphi_{i}\right)}-{\left(\frac{\sum_{i=1}^{I}P(\varphi_{i})\times \varphi_{i}}{\sum_{i=1}^{I}P(\varphi_{i})}\right)}^{2}},$$
where $\theta_{i}$ denotes the azimuth angle of arrival (AoA), and $\varphi_{i}$ represents the corresponding elevation angles. The azimuth angular of arrival spread (AAS) $\sigma_{\theta }$ and the elevation angular of arrival spread (EAS) $\sigma_{\varphi }$ are given.

#### 3.2.5. Channel Capacity

Channel capacity usually refers to the maximum information rate that a channel can transmit without errors. It is typically of practical significance in UAV-assisted emergency communications. Thus, it can be expressed as
(10)$$C=B\times \log_{2}\left(1+\rho \;\times \;|\sum_{i=1}^{I}\alpha_{i}\times e^{j\phi_{i}}{|}^{2}\right),$$
where ρ is the signal-to-noise ratio (SNR), and *B* is signal bandwith.

## 4. Numerical Results and Analysis

### 4.1. Measurement Verification

To validate the availability of the RT model and simulation data, UAV channel measurements were conducted in the mountainous scenarios. The measurement campaigns were conducted near the Xinglong Mountain Campus of Shandong University, around 36.5931° N and 117.0304° E. The location was relatively safe and representative. In addition, the channel measurements were taken in September, when the vegetation was relatively lush. Figure 4a shows the actual measurement scenarios.

The measurement system consisted of the air end and the ground end, as shown in Figure 4b,c. The air end consisted of the DJI M600 pro UAV, the Universal Software Radio Peripheral (USRP) X310, and the discone-type antenna. Due to the large weight of the mmWave signal transmitter and the maximum UAV load at 8.0 kg of weight, mmWave channel measurements were not conducted. The USRP X310 was used to transmit the pseudo-noise (PN) sequence. The antenna was deployed below the UAV propeller. The USRP X300 was used to receive the PN sequence, and signals were stored in the computer at the ground end. The receive and transmit gains of the USRPs were set to 30 dB. In addition, the sample rate was selected as 100 MHz, and the frequency was set to 3.6 GHz with a 80 MHz bandwidth. The horizontal flight speed was 5 m/s at altitudes of 40 m and 50 m from the Rx. Also, to obtain more accurate channel characteristics, the effects of the device, antennas, and cable were eliminated by back-to-back calibration [39] in data pre-processing. The CIRs were obtained from the calibrated data and calculated from
(11)$$h\left(\tau \right)=\mathrm{IFFT}\left(H\left(f\right)\right)=\mathrm{IFFT}\left(X_{\mathrm{rx}}\left(f\right)/X_{\mathrm{tx}}\left(f\right)\right),$$
where $X_{\mathrm{tx}}\left(f\right)$ and $X_{\mathrm{rx}}\left(f\right)$ are the frequency domain responses of the transmitted and received signals, respectively.

Based on the acquired CIRs h(τ), the MPCs can be extracted by the peak search algorithm to further analyze the channel characteristics. The peak search power threshold is chosen as the maximum value between the maximum power minus 20 dB and the average noise floor plus 5 dB [15]. As an example, Figure 5 shows the results of MPC extraction for the 93rd point of horizontal flight at a vertical altitude of 40 m. The threshold in the figure is −53.30 dBm, and channel data below the threshold is considered to be noise. The value of extracted MPCs is the received power, and the corresponding time point is considered as the delay. The first arrived path is seen as the LoS path, which has the minimum delay. Here, the delay of LoS path was 0.10 μs. In addition, the delays and powers of the corresponding NLoS paths could be further obtained.

Using exactly the same frequency, bandwidth, and flight speed as the real measurements, the propagation environment in WI software was reconstructed. The Rician K-factor was calculated from Equation (Equation 6). Figure 6 shows the cumulative distribution functions (CDFs) of the Rician K-factor for the RT data and measurement data. When the flight height was 40 m, the average values of the Rician K-factor for the RT data and measured data were 18.32 and 18.23, respectively. When the flight height was 50 m, the average values were 17.33 and 17.80, respectively. The CDF curves of the measured and simulated data were well fitted intuitively. Besides, to make the data more convincing, the Kolmogorov–Smirnov (K-S) test was employed to measure its accuracy and effectiveness. The K-S distance test value is typically utilized to compare if two distinct data sets stem from the same underlying distribution. The results show that the K-S test distance values (D values) for flight altitudes of 40 m and 50 m were 0.1530 and 0.1595, respectively. Through the comparison, we can see that the K-S test D values were below 0.21 (the upper quantile of 95% significant level) [40]. Hence, it can be assumed that the RT data and measurement data followed the same distribution. Therefore, the accuracy and practicality of the RT method and simulation results could be well demonstrated.

### 4.2. Different UAV Flight Positions

The location of the mountain and the distance from the Rx have a significant effect on the channel characteristics. The UAV was placed on the left and right sides of the Rx. The detail is shown in Figure 3a. In the real scenarios, the right side was a higher mountain. Figure 7 shows the PDPs at different positions. The birth and death of the scattering clusters at different times can be observed. The continuum shift variation of the MPC delay and power can be clearly seen in the figure. There were fewer multipath clusters in the mountainous scenarios compared to the typical urban scenes [27]. MB components in mountainous scenarios were less, and their existence times were short. Both Figure 7b,c have scattered paths at about 3.2 μs, but with different death times. Furthermore, the terrain surrounding the Rx in mountainous regions exhibited an uneven architecture, which gave rise to a significant and intricate ground diffuse reflection effect.

Figure 8 shows the CDFs of the RMS DS and Rician K-factor and the fitted curve using a Gaussian stochastic process. When the UAV was 150 m to the right of the Rx at a horizontal distance, it was greatly influenced by the mountain. The scattering path shared more energy, and the LoS path accounted for a relatively small percentage. Consequently, the average of the RMS DS was large, and the average of the Rician K-factor was small. For the Rician K-factor, the variance was small when the UAV was close to the mountain, thus indicating that the fluctuation was small with the movement of the UAV. In addition, the Rician K-factor was almost identical for the left and right 50 m. It is evident that the proximity to the mountain exerted a significant influence, thereby resulting in a richer scattering path and greater losses, which ultimately led to a decline in communications quality. Meanwhile, our empirical findings demonstrated conformity with our experimental expectations, which can be deduced from the data to draw the conclusion.

### 4.3. Different UAV Flight Trajectories and Altitudes

UAV always fly with a 3D arbitrary trajectory and different flight altitudes. After determining the start and end positions of the UAV flight, there are often three types of flight: hovering flight (trajectory 1), point-to-point direct oblique flight (trajectory 2), and vertical flight to the end point altitude, followed by horizontal flight to the end point (trajectory 3). The trajectories are shown in Figure 3b. The trend of the PDPs at different trajectories can be observed in Figure 9. In Figure 9a,b, a clear difference between the two trajectories, hovering and straight, can be evidently observed. In Figure 9c, the turning point where the UAV changes from vertical to horizontal flight can be seen at 15 s. The delays of the LoS trajectories at the start about 0.6 μs and end points about 1.2 μs of the three figures were the same but at other times were different, thus verifying that the location points at the start and end points were the same, but the routes were different.

Figure 10a reflects the CDF of the RMS DS at different altitudes at 100 m, 150 m, and 200 m. It was found that the RMS DS of the UAV channel was smaller at low altitudes and larger at high altitudes. The main reason is that at high flight altitudes, the UAV tends to be on the middle and upper sides of the mountain, where it can observe a larger number of scatter signals and is more influenced by the mountain. This phenomenon is consistent with both the GBSM model and the RT data for UAV communications in the urban scenarios, but it is in contrast to the rural scenarios [20,27]. Also, the RMS DS was smaller for mountain scenes compared to urban scenes at the 28 GHz frequency band (urban scenes: mean value 0.55 μs at an altitude of 100 m, and mean value 0.6 μs at an altitude of 150 m in [20]). Figure 10b shows the CDF of the RMS DS for different trajectories. The UAV channel delay dispersion was most obvious in trajectory 3, and the mean values of the RMS DS were basically the same for trajectory 1 and trajectory 2. The possible reason is that trajectory 3 was at a relatively high position for a long time, and this effect of flight altitude on the DS has been illustrated in Figure 10a. Therefore, its statistical characteristics in Figure 10b accounted for much of the large DS due to high altitude. This needs to be taken into account for future flight trajectory design.

Figure 11 shows the CDFs of the channel capacity at different altitudes. In complicated mountainous scenarios, the channel capacity is usually low due to multipath propagation and PL. It can also be seen that the average channel capacity decreased significantly as the UAV rose. The average channel capacity was about 1.22 Gbps at a vertical altitude of 100 m and 177 Mbps at 200 m. Furthermore, the values of the channel capacity accounting for 80% were 1.77 Gbps, 0.76 Gbps, and 0.29 Gbps, which corresponded to the heights of 100 m, 150 m, and 200 m, respectively. This is because the higher flight altitude results in signal fading, increased noise, and extended transmission delays in mountainous scenarios. This ultimately leads to a diminished effective channel capacity.

The received power at different locations in UAV-assisted emergency communications is a very important indicator. The 3721 Rx points were deployed in a ground area of 1.5 km^2^ using a sampling interval of 25 m. The UAV was positioned at a location approximately around point (25, 15), assuming that the first Rx located at the bottom-right corner was the origin (0, 0). The coverage range increased significantly with an increase in altitude, which is reported in the form of Figure 12. The UAV achieved a coverage range limited to solely the sides of the valley at 100 m in altitude. The signal strength was relatively weak at 150 m in height due to significant shadowing caused by the back of the mountain, which led to only partial coverage of the mountain. At 200 m, partial coverage of the rear of the mountain was achieved, and improvements in energy resources were noted.

### 4.4. Different UAV Carrier Frequency Bands

The selection of the frequency bands often has an impact on the propagation channel. The choice of frequency bands in 6G UAV-assisted emergency communications is still undetermined. Therefore, in this paper, the frequency bands 3.5 GHz, 4.9 GHz, 28 GHz, and 38 GHz were selected, and the bandwidths were set to 80 MHz, 80 MHz, 1 GHz, and 1 GHz, respectively. As an example, the PDP at the flight time of 10 s at different frequency bands was selected, as shown in Figure 13. The summation power of the MPCs could be calculated as −42.46 dBm, −45.89 dBm, −59.44 dBm, and −62.37 dBm. The LoS delay was basically the same, while the power was significantly different between four different frequency bands. The mmWave attenuation was significantly greater than sub-6 GHz and tended to become greater with increasing frequency. The reflected and scattered multipaths in each band had similar time delay distributions overall, but there were some differences. For example, the multipath at the 3.2 μs time delay was observed only at 28 GHz and 38 GHz. In addition, due to the large bandwidth of mmWave and the high time delay resolution, the mmWave was susceptible to the influence of mountains, and the scattering phenomenon was more obvious.

Figure 14 shows the PL scatter plot and the fitted lines with the FI model for different positions. Obviously, it had a higher loss in mmWave frequency bands, which reached 105–122 dB. For sub-6 GHz frequency bands, it had a loss of 85–102 dB. In addition, the PLE *n* were basically similar for the four frequency bands, and the intercept ν shows a positive correlation with frequency. The variance of the SF σ was essentially around 1–2 dB.

Figure 15 shows the CDFs of the RMS DS, AAS, EAS, and Rician K-factor at different frequency bands. All data were fitted using a Gaussian normal distribution N(μ, $\sigma^{2}$), and the mean and variance parameters are shown in Table 3. It can be seen that the 38 GHz channel had the largest mean value of the RMS DS. The mean value of the AAS at 28 GHz was small, and the Rician K-factor was large. For the EAS, the variance of these four bands was essentially similar, and 3.5 GHz had the largest mean value.

Figure 16 shows the channel capacity at different frequency bands. The 3D straight-line distance of the Tx and Rx ranged from 113 m to 622 m, and the UAV flight trajectory details are shown in Figure 3c. At 28 GHz and 38 GHz, both the CIR complex matrix and SNR were small at long distances due to the large attenuation in mountainous scenarios. Therefore, the channel capacity was reduced for 3D distances greater than 400 m, even when multiplying by a wider channel bandwidth *B*. When the distance was less than 250 m, the mmWave channel capacity increased dramatically, up to 4.1 Gbps. Due to the greater attenuation at 38 GHz, the 28 GHz frequency band could provide higher channel capacity than the 38 GHz band overall. When the 3D distance from the Tx to Rx exceeded 400 m, the attenuation of the mmWave was too large to provide stable communications, and the sub-6 GHz channel capacity fluctuated less. In general, when the 3D distance is below 200 m, the mmWave can provide better communication services during emergency communications in mountainous scenarios. In the future, massive multiple aircraft mmWave antennas can be used to obtain diversity gain, and suitable modulation and coding schemes can be used to improve the data transmission rate and interference immunity.

The above UAV AG channel characteristics were analyzed considering sunny weather. During emergency situations, there are usually adverse environments, including rainy weather and severe thunderstorms. It is necessary to fully consider the impact of precipitation on the channel characteristics. In the ITU-R P. 838-3 standard [41], the specific attenuation $\gamma_{R}$ is predicted as:(12)$$\gamma_{R}\left(\mathrm{dB}/\mathrm{km}\right)=k\times R^{\alpha },$$
where *k* and α are frequency-dependent correlation coefficients. *R* is the rainfall rate in mm/h. In this paper, the atmospheric pressure was adjusted to 913 mbar, the temperature to 12.5 °C, and the relative humidity to 90% to change the atmospheric composition. The physical quantities were derived to be closer to the situation affected by these climate factors after-processing the output data. The attenuation of *n*-th ray is calculated by
(13)$$A_{Rn}\left(\mathrm{dB}\right)=\gamma_{R}\times \kappa \times d_{n},$$
where $d_{n}$ is the actual path length, and κ is distance factor. The detailed calculation can be found in the ITU-R P. 530-17 standard [42]. The meteorological attenuation and Gaussian random phase $\varphi_{A}$ were introduced [43]. The received power is
(14)$$P_{A}^{{}^{\prime }}=20\log_{10}\left|\sum_{n=1}^{N}10^{\frac{P_{n}-A_{Rn}}{20}}\times e^{j\left(\varphi_{n}+\varphi_{A}\right)}\right|.$$

In order to reduce the effect of the error, the average value of the received power for the whole flight trajectory was calculated. Figure 17 shows the simulation results. The rainfall rate *R* was set from 1 mm/h to 50 mm/h. It can be observed that the UAV AG mmWave channel was easily affected by rainwater and atmospheric components. For 3.5 GHz and 4.9 GHz, the received power hardly declined when the rainfall rate was 1 mm/h to 10 mm/h. However, it declined by 1.31 dB at 28 GHz and 1.91 dB at 38 GHz. For a larger rain rate, some fluctuations were observed at 3.5 GHz and 4.9 GHz, possibly due to random phase effects. In terms of the mmWave channel, the signal propagation attenuation significantly increased with the rainfall rate, especially at 38 GHz, where the maximum attenuation reached 9.15 dB. Therefore, in UAV-assisted emergency communication under extreme weather conditions, it is necessary to consider the impact of precipitation on the channel.

## 5. Conclusions

UAV-assisted emergency communications in complicated mountainous scenarios have practical significance in future 6G communications. In this paper, the RT method was applied to obtain the UAV AG channel data in mountainous regions and verified by the real channel measurements. Based on the acquired channel data, the channel characteristics with different flight positions, altitudes, and trajectories were analyzed. Results have shown that the positions of the UAV in relation to the mountains have an impact on the scattering clusters evolution. The PDPs and delay extensions fluctuated with different flight trajectories. The altitudes of the UAV flights could significantly affect the signal coverage area. The higher the altitude of the flight, the wider the signal coverage areas were. However, this also led to the increase of the RMS DS and the decrease of the channel capacity. Furthermore, the channel characteristics of different frequency bands were also compared and analyzed. In addition, the study of mmWave propagation channels was provided, which presented that mmWave characteristics were easily affected by the mountainous environment. The mmWave transmission was susceptible to rain attenuation, while the sub-6 GHz reflected a certain degree of stability. In the future, MIMO and mmWave channel measurements will be conducted, and channel parameter calculation will be optimized in order to promote a comprehensive analysis of the multi-dimensional UAV communication characteristics.

## Figures and Tables

**Figure 1 sensors-23-04998-f001:**
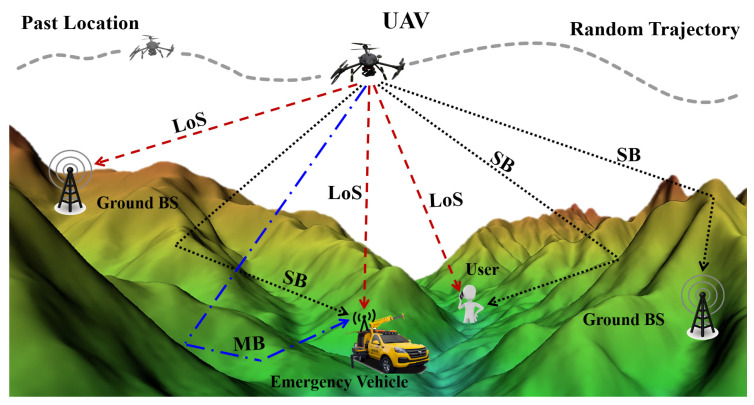
Architecture of UAV AG propagation model in complicated mountainous scenarios.

**Figure 2 sensors-23-04998-f002:**
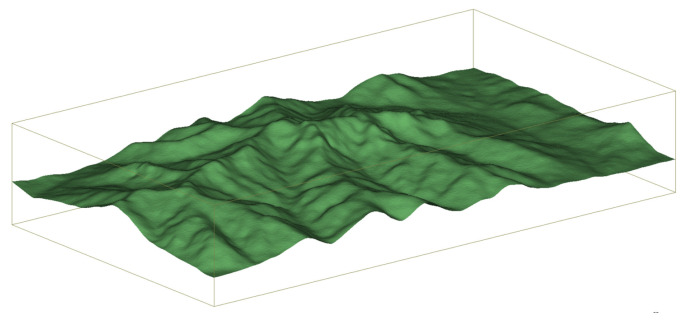
3D simulation terrain environment for the mountainous scenarios.

**Figure 3 sensors-23-04998-f003:**
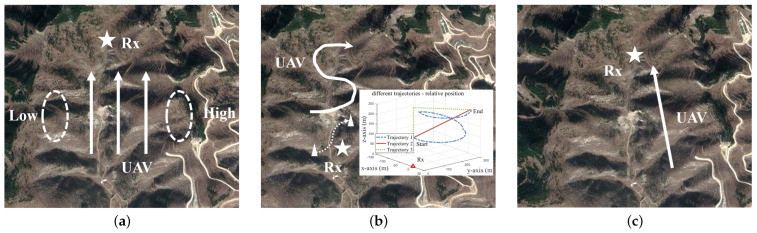
Satellite view and flight detail sketch of research on (**a**) UAV flight positions, (**b**) UAV flight trajectories and altitudes, and (**c**) UAV carrier frequency and environmental factors (The arrow represents the flight direction, the circle represents the approximately location of the mountain, the triangle represents the starting and ending points of different trajectories, and the star represents the Rx).

**Figure 4 sensors-23-04998-f004:**
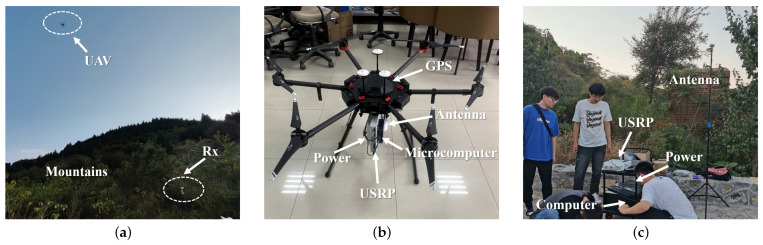
(**a**) Channel measurement environment, (**b**) UAV transmitter, and (**c**) ground receiver in mountainous scenarios.

**Figure 5 sensors-23-04998-f005:**
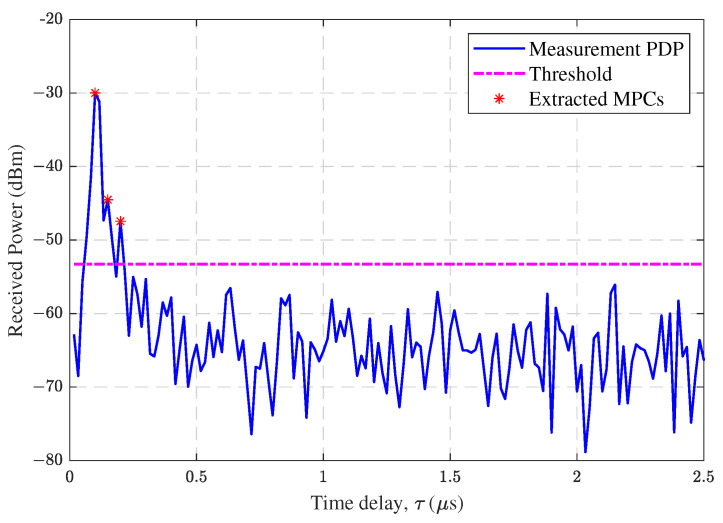
Measurement PDP and extracted MPCs at point 93.

**Figure 6 sensors-23-04998-f006:**
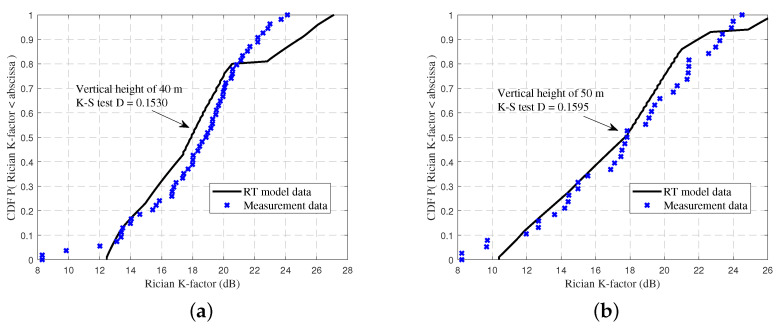
The CDFs of Rician K-factor at altitudes of (**a**) 40 m and (**b**) 50 m for the RT data and measurement data.

**Figure 7 sensors-23-04998-f007:**
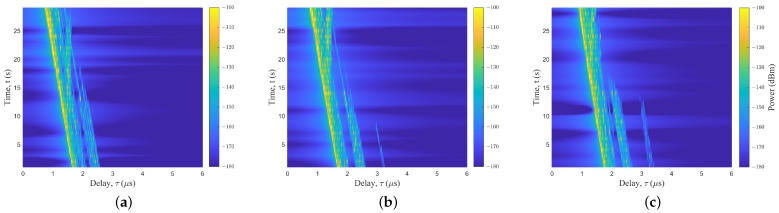
The PDPs of UAV flight positions at (**a**) left—50 m, (**b**) right—50 m, and (**c**) right—150 m.

**Figure 8 sensors-23-04998-f008:**
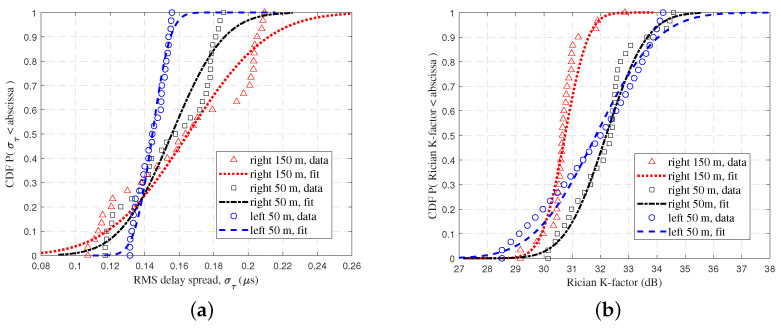
The CDFs of (**a**) RMS DS and (**b**) Rician K-factor at different UAV flight positions.

**Figure 9 sensors-23-04998-f009:**
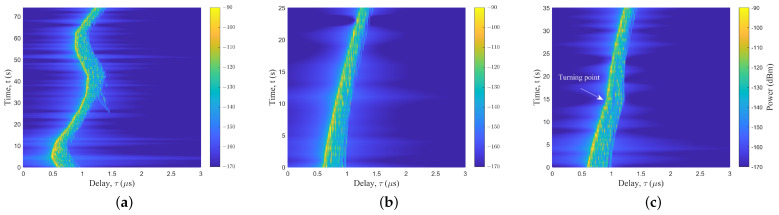
The PDPs of UAV flight with (**a**) trajectory 1, (**b**) trajectory 2, and (**c**) trajectory 3.

**Figure 10 sensors-23-04998-f010:**
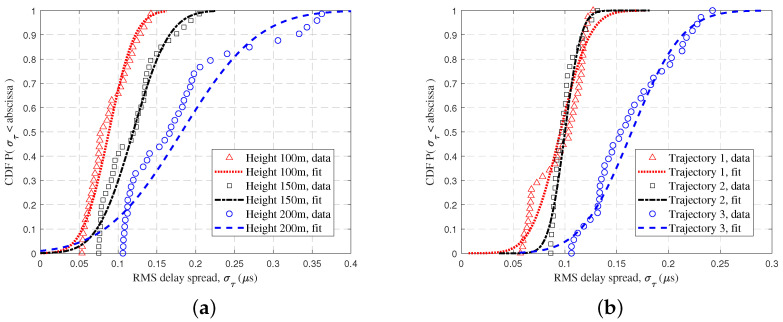
The CDFs of RMS DS for UAV flight with (**a**) different flight altitudes and (**b**) different flight trajectories.

**Figure 11 sensors-23-04998-f011:**
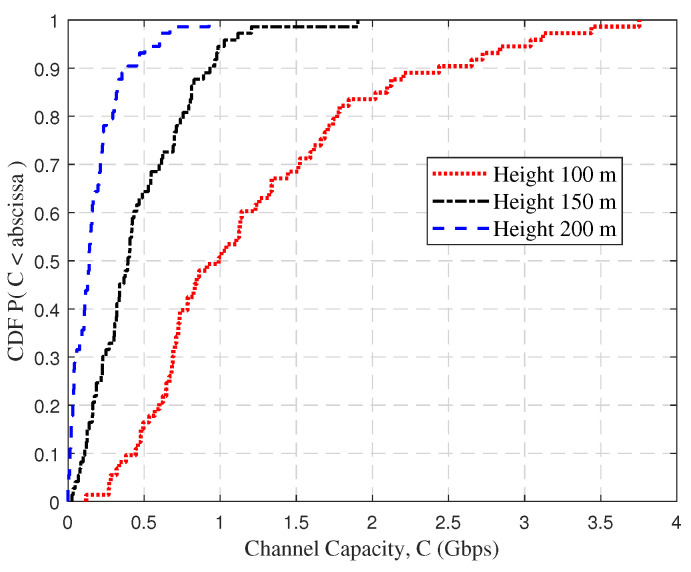
The CDFs of channel capacity at different UAV flight heights.

**Figure 12 sensors-23-04998-f012:**
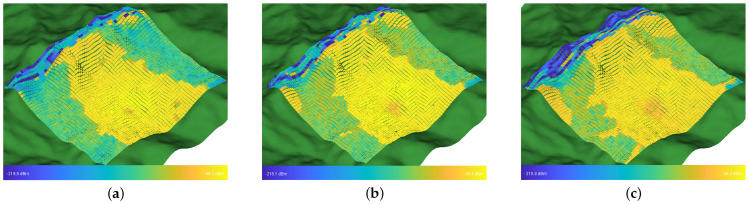
Ground end received power at UAV flight altitudes of (**a**) 100 m, (**b**) 150 m, and (**c**) 200 m.

**Figure 13 sensors-23-04998-f013:**
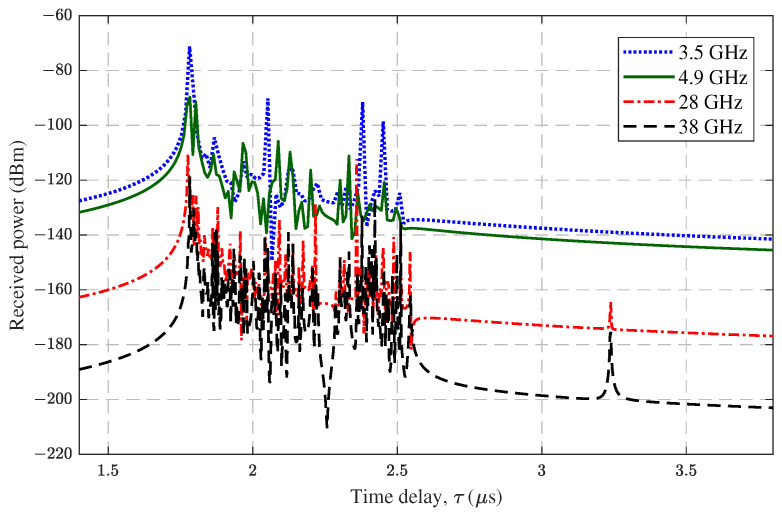
The PDPs of different UAV carrier frequency bands (flight time = 10 s).

**Figure 14 sensors-23-04998-f014:**
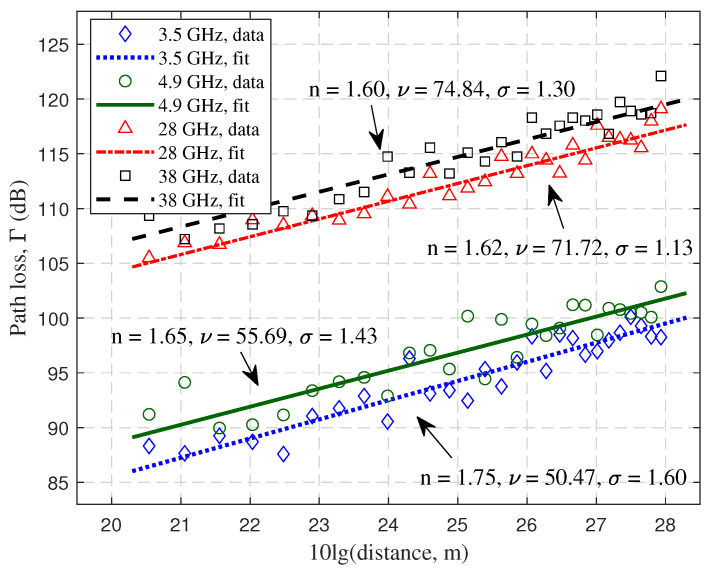
The PL and fitted lines based on FI model at different frequency bands.

**Figure 15 sensors-23-04998-f015:**
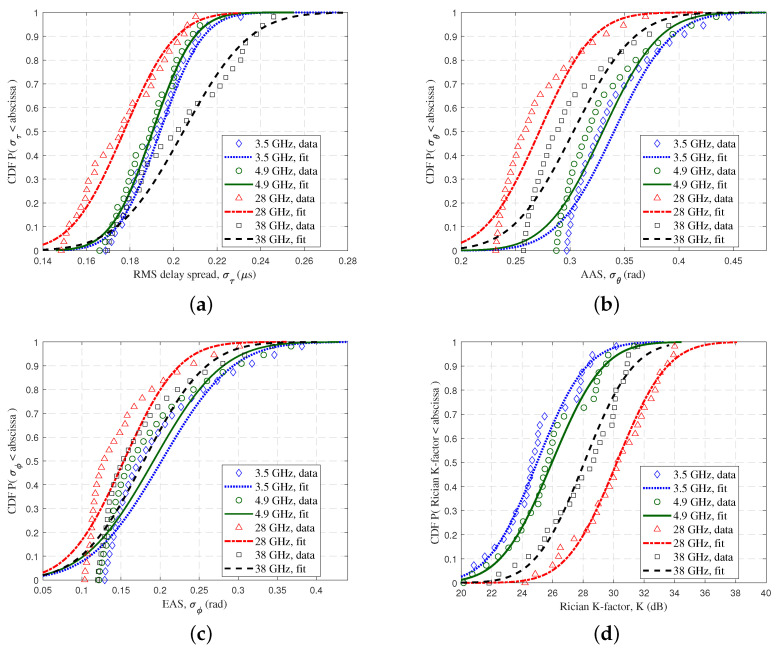
The CDFs of (**a**) RMS DS, (**b**) AAS, (**c**) EAS, and (**d**) Rician K-factor at different frequency bands.

**Figure 16 sensors-23-04998-f016:**
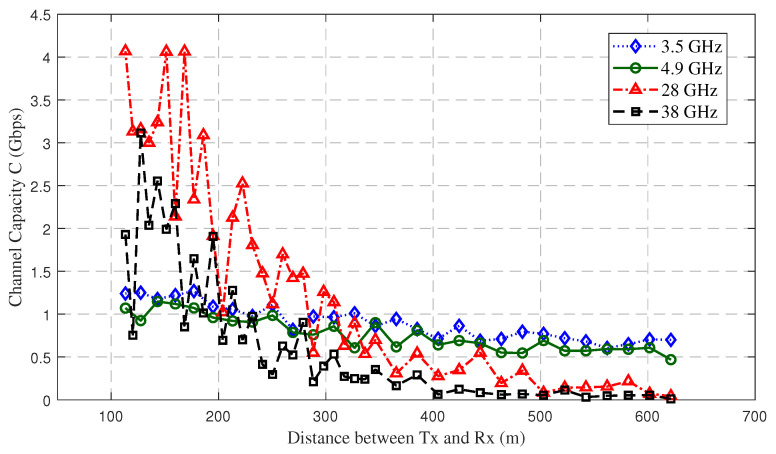
Channel capacity of different UAV carrier frequency bands.

**Figure 17 sensors-23-04998-f017:**
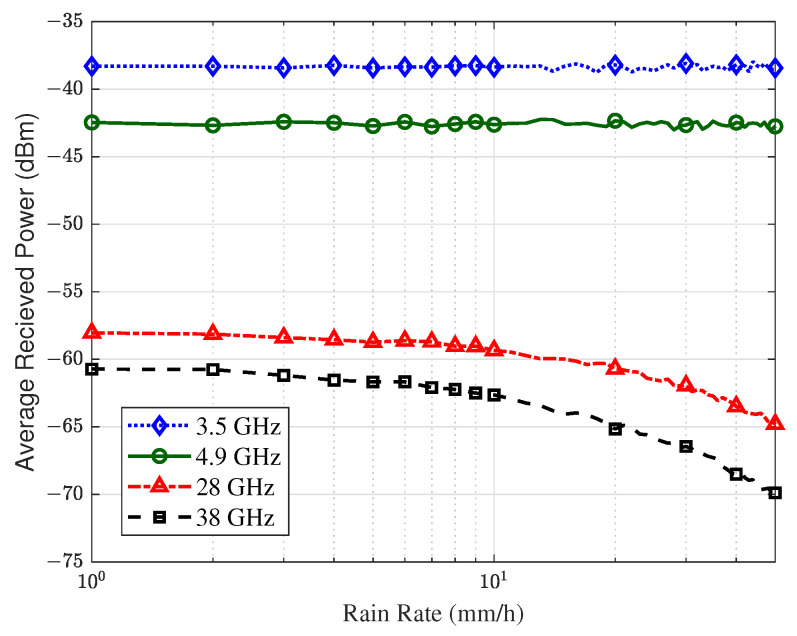
The relationship between received power and rain rate at different frequency bands.

**Table 1 sensors-23-04998-t001:** Parameters of soil materials at different frequency bands.

	3.5 GHz	4.9 GHz	28 GHz	38 GHz
**Permittivity**	18.18	15.76	5.70	4.80
**Conductivity (S/m)**	0.76	1.22	9.50	22.00
**Thickness (m)**	0.00	0.00	0.00	0.00

**Table 2 sensors-23-04998-t002:** Details of simulation propagation parameters.

Research Subjects	I	II	III
**Frequency**	28 GHz	28 GHz	-
**Bandwidth**	1000 MHz	1000 MHz	-
**Transmit Power**	40 dBm	40 dBm	40 dBm
**Flight Altitude**	65 m	-	75 m
**Flight Trajectory**	Straight line	-	Straight line
**Flight Speed**	10 m/s	10 m/s	10 m/s
**Flight Distance**	300 m	-	550 m
**Antenna Pattern**	Omnidirectional	Omnidirectional	Omnidirectional
**Antenna Max Gain**	8 dBi	8 dBi	8 dBi
**Waveform**	Sinusoid	Sinusoid	Sinusoid
**Polarization**	Vertical	Vertical	Vertical
**Ray Spacing**	0.25 Ω	0.25 Ω	0.25 Ω
**Reflection/Diffraction** **/Transmission**	6/1/0	6/1/0	6/1/0

**Table 3 sensors-23-04998-t003:** Parameters of normal distribution at four frequency bands.

	3.5 GHz	4.9 GHz	28 GHz	38 GHz
**RMS DS (μs)**	N (0.1938, 0.0156${}^{2}$)	N (0.1904, 0.0152${}^{2}$)	N (0.1772, 0.0190${}^{2}$)	N(0.2045, 0.0240${}^{2}$)
**AAS (rad)**	N (0.3403, 0.0416${}^{2}$)	N (0.3300, 0.0411${}^{2}$)	N (0.2717, 0.0390${}^{2}$)	N(0.3030, 0.0443${}^{2}$)
**EAS (rad)**	N (0.2018, 0.0718${}^{2}$)	N (0.1914, 0.0698${}^{2}$)	N (0.1536, 0.0556${}^{2}$)	N (0.1791, 0.0630${}^{2}$)
**Rician K-factor (dB)**	N (24.9784, 2.5552${}^{2}$)	N (25.9556, 2.6659${}^{2}$)	N (30.2080, 2.5964${}^{2}$)	N (28.0750, 2.5295${}^{2}$)

## Data Availability

Not applicable.

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
