# Peer review of "Channel Characterization and Modeling for 6G UAV-Assisted Emergency Communications in Complicated Mountainous Scenarios"

_sensors, 2023, doi:10.3390/s23114998_

Round 1

Reviewer 1 Report

Please see the attached comments.

Minor editing of English language is required.

Reviewer 2 Report

The paper presents a measurement campaign of UAV communications in mountainous scenarios. Multiple frequency bands (including mmWave) have been used, which is a relevant technology for future communication systems.
While the general system setup is described clearly, the presentation of the results and conclusions needs to be improved.

1. The main part of the paper lists the results of the measurements (and their analysis of derived quantities). However, the discussion of the measurement data and the resulting conclusions is very short and too vague.

2. On page 11, it is said that "Figure 11 shows the channel capacity at different altitudes." That is not correct. The figure in the presented draft shows the distribution function of the channel capacity. Therefore, all of the following descriptions of the figure need to take this statistical nature into account, e.g., by explaining that the average channel capacity is higher at lower flight altitudes. Furthermore, the sentence "The channel capacity is about 1.8 Gbps at a vertical altitude of 100m, and 300 Mbps at 200m" does not make sense for this figure, since Figure 11 explicitly shows that a range of channel capacities are possible for all altitudes.

3. The quality of the figures (especially the choice of colors) needs to be improved. The annotations in Figure 3 and Figure 4 are hard to read against the background. This gets even worse when the paper is printed in grayscale.

4. Similarly, the jet or rainbow color map should never be used at all. A perceptually uniform color map should be used instead. (Figure 7, Figure 9, Figure 12)

5. The colors and line styles in all plots should be unified, e.g., a dotted blue line is used for 3.5 GHz in Figure 13 while a bright green is used in Figure 15. Additionally, the bright green is very difficult to see.

There are some typos in the figures, e.g., "Poewr" in Figure 5 and "bps/Hz" in Figure 11 should probably be "Gbps/Hz".

In general, there are a lot of language mistakes (missing "the", ...).

Round 2

Reviewer 2 Report

The authors have addressed all of my comments. Especially the presentation of the figures is much clearer now.

---